# Reproducibility report for "On Disentangling Spoof Trace for Generic Face Anti-Spoofing"

## Reproducibility Summary

**Scope of Reproducibility**

From previous research(1), it has been demonstrated that the face anti-spoofing problem can be treated as a denoising problem where spoof images can be disentangled into two parts – the live counterpart and the noise. Based on that, this paper(2) suggests that the disentanglement will be more appropriate if done in four parts - s, b, C, T where - s, b represent color range bias and balance bias, C denotes the smooth content patterns and, T is the high-level texture patterns. They claim that by leveraging these features, one can find the live counterpart of any image as well as the spoof counterpart of the live inputs by warping. We identify the main scope of reproducibility is to verify whether it is possible to generate the live counterpart from any image by subtracting the spoof trace.

**Methodology**

We approach this challenge in the following manner. Firstly, we have reproduced the paper in PyTorch while taking help from the paper, the authors, and the official implementation. This process took us around a month. Secondly, we checked the soundness of our PyTorch implementation by taking the same input in both our and their implementation. Later, we train on MSU SiW Protocol-1 and OULU NPU Protocol-1 to verify whether we have succeeded to reproduce their results in PyTorch. Each epoch on a single V100 took around 12 hours. Finally, we went on to propose several improvements over a few identified limitations of the original paper.

**Results**

While verifying the reproducibility, we outperformed the result of OULU NPU Protocol-1 as we got an ACER of $1.195\%$ compared to their $1.9\%$. For MSU-SiW Protocol-1, we achieved an ACER of $0.53\%$ while they found ACER of $0\%$. Later, we verified whether it is possible to get a perfect live counterpart from any input image. We concluded that it is not always possible to get perfectly warped spoof images in many cases. Later, we proposed a few techniques to improve the generation of the live counterpart based on our observation.

**What was easy**

Although the method was really complicated, as the official implementation was easy to follow, we were at ease while re-implementing the architecture of the paper in PyTorch.

**What was difficult**

The hyperparameters for producing the metrics were not given and, as mentioned in the paper, the authors selected them by brute-forcing. We found it really time-consuming to find the best result as the method is prone to hyperparameters.

**Communication with original authors**

After we read the paper and started implementing the code, we had a few confusions for which we mailed the author and got help from them. Later, we suggested an inconsistency between the paper and the implementation.

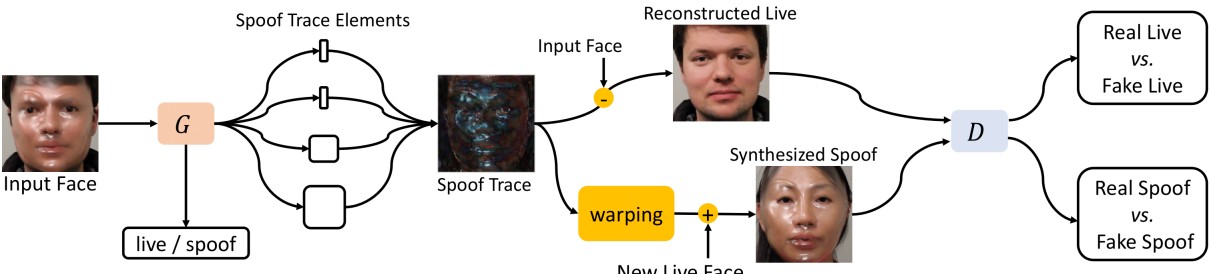

Figure 1: Summary of the whole process of the paper[1]

## 1    Introduction

In recent years, the tendency to depend on biometric authentication is increasing. Of all bio-metric authentication technologies, face recognition is one of the most popular technologies. One of the main reasons for the widespread use of this system is its non-intrusive nature of acquisition and ease of use(3). However, with the extensive use of face recognition brings into the question of the security of face-authentication systems and its robustness against various forms of malicious attacks. Sadly, these systems are susceptible to various forms of presentation attacks (PA), or more commonly known as spoofing. Mere printouts or recorded videos of faces presented in front of the biometric sensor are enough to fool many face-recognition systems.

In the past, hand-crafted features(4; 5; 6; 7; 8) like HOG and LBP were used to tackle the problem of face anti-spoofing. However, with the revolution of high quality cameras, these techniques have become ineffective. In recent years, CNNs have been adopted(9; 10; 11; 12) as the preferred solution for this problem. CNNs, with techniques like depth supervision or pixel-wise binary supervision(10; 13), have shown great potential. The paper(2) we reproduced proposes a Generative Adversarial Network based technique to find spoof traces from an image. It is based upon the assumption that, spoof attack is an additive noise of the live face which is shown behind the spoof attack. This technique has shown great promise as the proposed method achieves state of the art (SOTA) performance in both inter and intra dataset testing across different research datasets.

## 2    Scope of reproducibility

Previously from (1), it has been seen that, face anti spoofing problem can be treated as a de-noising problem. Based on this finding, the paper we reproduced has extrapolated the following claims:

- Claims that the equation-2 will produce live counterpart of any image where, $I$ is the input image, and $s, b, C, T$ of 1 are the spoof trace components. Here, the right hand side of equation-2 forms the spoof trace of the input image, $I$. $I_{spoof\_trace}$ should be zero if the input is a live input. Finally,  is expected to be the live counterpart of the input image $I$.
- Treats several cues as irrelevant and claims that it hinders the generalization.
- Claims that hard and rare(e.g., 3D face data) samples can be created using the spoof trace.

$$I_{spoof\_trace} = (1 - s) * I + b + C + T \tag{1}$$

$$\hat{I} = I - I_{spoof\_trace} \tag{2}$$

## 3    Methodology

Based on the claims from Section-2, we took the following approaches to verify their claim and then go on to propose ways to tackle few problems that we found in their paper.

---

[1]This figure is taken from the official repository of the paper

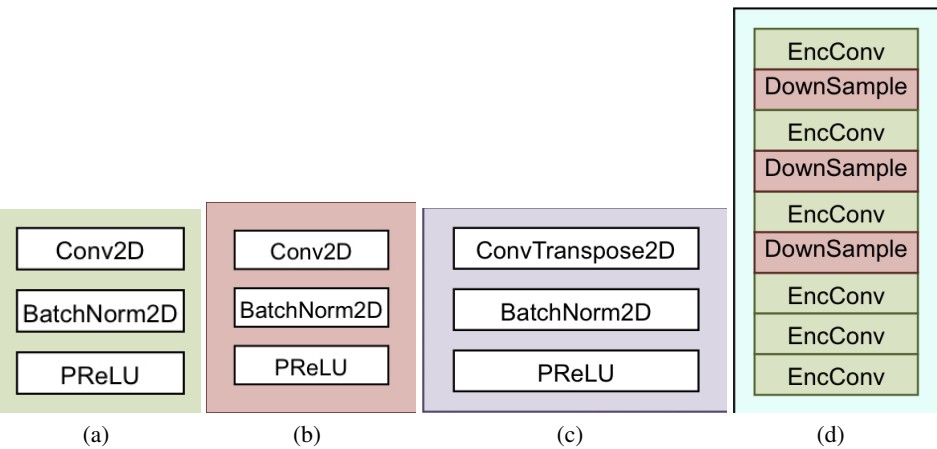

Figure 2: (a) Stack of layers used after each Convolutional layer, (b) Stack of layers used for downsampling, (c) Stack of layers for upsampling, (d) The discriminator used in the Multi-discriminator

- Equation-1 is used to get the spoof trace of any input image. Later on, equation-2 was proposed to get the live counterpart of any given input. We follow the equations and verify whether this claim is true or not. Our result can be seen in figure-5.
- They claim that spoof trace helps to remove the outside cues which effect the generalization ability of the model. We use Fast Fourier Transformation based features as outside cues to verify the claim. We have attached figure-4 to show the outcome of our experiment.
- Finally, while demystifying the above claims, we observed a limitation which is – the reconstructed live counterpart of spoof images take up a different colored hue which can be seen in figure-5. We explored two ways to solve this problem. The results of the approaches can be seen in [6, 7].

## 3.1 Experimental setup

### 3.1.1 Architecture

While implementing the research, we followed the architecture they shared in the paper. However, we changed few small changes after discussing with the authors. Few things that are to be noted – the batch size should always be even numbered as we are required to take equal number of spoof and live input images for each batch. One other thing that we were informed about was that they used $PReLU$ instead of $LeakyReLU$ afterwards as they found better results with the former. Besides, input sizes for the multi-scale discriminator are also changed from $(256, 128, 64)$ to $(256, 160, 40)$. Because, according to the authors, the later one gave better results.

Apart from these changes, we kept every other thing similar to the official implementation. We sent equal number of spoof and live images into the generator – which outputs the probabilities of early spoof regression and the disentangled spoof traces $s, b, C, T$. Once we find the disentangled values, we create the spoof trace using Equation-1 and then, we get the live counterparts of the inputs using Equation-2 which we call Reconstructed Live image. Later, we warp the spoof trace using (14). We use the warped spoof trace to create spoof counterpart of the live inputs. Finally, we send all the actual inputs and their synthetic counterparts to the discriminator. From Figure-1, the whole process of the network can be seen.

In figure-3, the network architecture is shown wheareas, the details of the common blocks of the architecture can be seen in figure-2.

### 3.1.2 Datasets

We used protocol-1 from **MSU-SiW** and **OULU-NPU** to verify our claims. Below, the datasets are discussed:

**OULU-NPU**

OULU-NPU (15) is one of the most recent datasets in the face anti-spoofing dataset family. It consists of 55 subjects and all the recordings were done using six phones and in three different environments (session). The session was

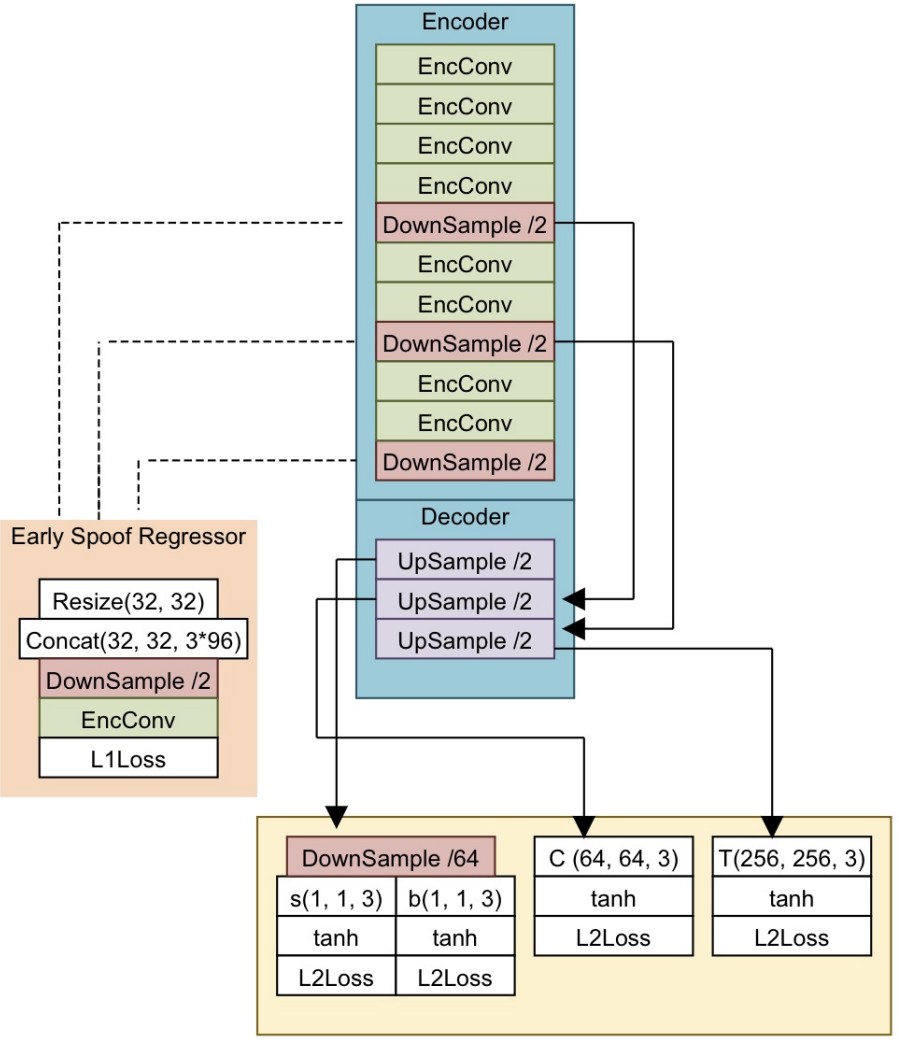

(a) Generator for Disentangling the spoof traces

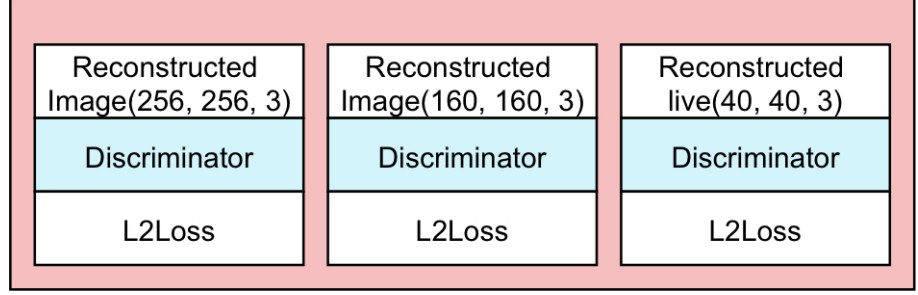

(b) Discriminators for the resized and reconstructed live counterpart of the spoof images.

Figure 3: The coloured blocks are referenced from Figue-2.

introduced to better simulate real world situations. To enforce generalization in the data, they followed "leave one out" approach where for training, five of six mobile phones are taken and the remaining one is used for testing. There were 3960 spoof videos (print and video replay combined) and 1980 bona fide videos. This makes it by far the most variant and definitely larger than MSU-MFSD (16) dataset. Also, there were 4 different protocols. The protocol names were set based on the level of difficulty. Following points show a brief description of OULU-NPU (15) protocols -

- Protocol I consists of a Test set which has different environments (session) than Dev and Train set.
- Protocol II ensures difference in instruments (smartphones) in the Test set as opposed to the Dev and Train set.
- Protocol III uses recording from different phones for Train and Test set. This helps to verify generalization capability of any network.
- Protocol IV is the hardest protocol of them all. It combines every constraints of the previous protocols. Also, with smaller subset of videos for training and evaluation.

**MSU-SiW**

The previous datasets before this (17) had lesser subjects compared to this one's 165 subjects where each subject has up to 8 bona-fide videos and 20 spoof videos. All the videos are captured in 30 frames per second on high definition Cameras – a Canon EOS T6 and a Logitech C920. The dataset provides print and replay attacks based on four sessions to capture the videos which included pose, illumination, distance and expression. Furthermore, to set a baseline for future studies, three protocols are created -

- **Protocol I** uses the first 60 frames of the videos for training and uses all the videos on the testing set. This would test if the model is generalized for contrasting face poses and expressions.
- **Protocol II** leverages the leave-one-out strategy to train on three mediums and test on the remaining one, to understand if the model is invariant to varying mediums.
- **Protocol III** tests the models performance on different forms of attack by training on one print attacks, testing on replay attacks and vice-versa.

### 3.1.3 Resources

We mostly followed the original paper and the github repository provided by the authors to reproduce our version. During the implementation phase, we discovered an inconsistency between the paper and the official implementation in the synthetic early spoof regressor loss. We even created a pull request to fix the problem although the authors fixed the problems on their own afterwards. The details of our pull request can be found here. Besides this, we observed few other inconsistencies in labels which were resolved by contacting the authors. Moreover, while implementing, we had immense support from PyTorch discuss forum.

### 3.1.4 Memory Usage

We had access to a single Tesla V100 and a single Tesla K80 for training. For verification purposes, we used the v100 as we required to train using batch size of 8 to be consistent with the original paper. For the separate extensions, we performed all our experiments on a single Tesla K80.

## 3.2 Implementation

We implemented most of our codes in **PyTorch**. The original implementation was done in $tensorflow - 1.13.0$. As discussed previously, to fit the spoof trace of the input image to another input, one requires warping the input image. While warping, the original implementation requires a method of $tensorflow$ named $gather\_nd$. This method takes a tensor and the indices(up to n-dimension) representing locations in that tensor. Then, it returns the elements of the tensor corresponding to the indices provided. However, till $PyTorch - 1.6$, this method is not implemented in PyTorch. Although, we implemented our version of that method in PyTorch, it was making the execution of our implementation too slow. So, instead of using our implementation, we adapted the tensorflow method in ours. We did this by converting the subsequent PyTorch tensor to a TensorFlow tensor for that specific block and later, converted it back to a PyTorch tensor. Apart from this, we implemented whole codebase from scratch while only looking into the official implementation for hyperparameters and architecture details. To verify that our implementation is same as theirs, we sent same input to both implementations. We found that, the output result matches up to $two$ decimal points. Our implementation can be found here: https://github.com/gazeai/STDN-PyTorch

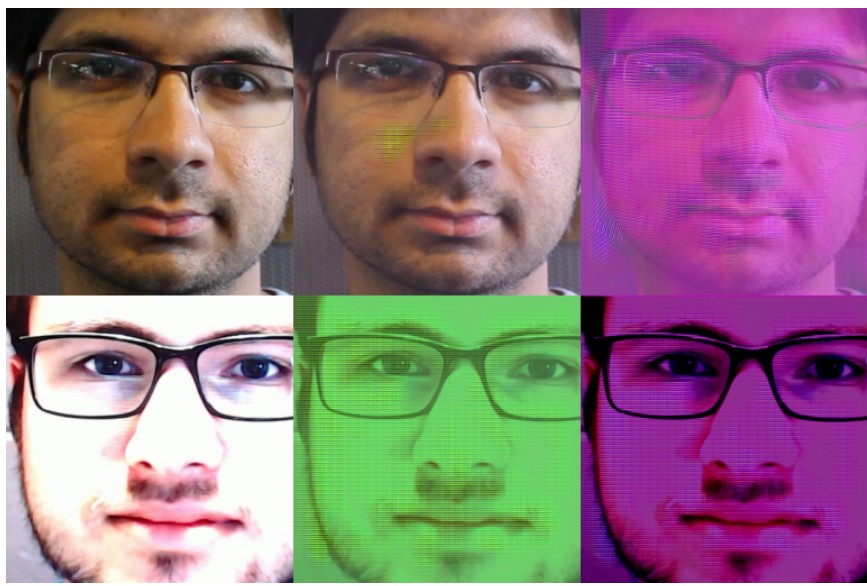

Figure 4: Generated outputs after using FFT as an outside cue. Row-1: (a) Live input, (b) Live counterpart of (a), (c) Spoof counterpart of (a) and, Row-2: (a) Spoof input, (b) Live counterpart of a, (c) Spoof trace of a

## 4 Our Contribution

### 4.1 Fast Fourier Transformation as Cue

As per their paper, cues of spoof are too varied in real life scenario and spoof detection as a binary classification task would take those cues into account to make the final decision. But in reality, those cues might be irrelevant to spoof. For example illumination. This hinders the generalization ability of a model. Often the illumination and environment present in a research dataset contains much less variations than a real life scenario. To deal with this, they use only spoof trace to make the final decision. This has shown to prove good generalization ability as it doesn't take those spoof cues into account.

We however, tried incorporating Fast Fourier Transformation (FFT) into the final decision making process. To add FFT, we created a new head in the $Encoder$ block of the generator. This head transforms the features using FFT and later, we added the FFT features with the final encoder output. This effected the early spoof regression features and thus, effecting the final prediction. In figure-4, we can see the results. It changes color tone to pink which is also visible in the main implementation of the paper[Figure-5]. However, it performs worse than the main implementation. Because, the FFT implementation introduces a checkerboard texture in the generated images which is not present in the main implementation.

### 4.2 MinMax Normalization

In Equation-2 , I is the input image within the range of $[0, 1]$ and, from equation-1, *s, b, C, T* ranging within [-1, 1] according to the official implementation of the paper. If we think of a scenerio where s=b=C=T=-1, the reconstructed image $I_{hat}$ will fall into the range of [4, 5] and on the other hand, if s=b=C=T=1, the range will be [-4, -3] which will produce unrealistic traces. This we can see in Figure-5 where the images add a pink and green hue in the faces and traces. To solve this problem, our initial solution was to normalize the reconstructed image $I_{hat}$ using equation-3 which ranges from $[0, 1]$.

$$I_{hat} = \frac{I_{hat} - \min I_{hat}}{\max I_{hat}} \tag{3}$$

From Figure-6, we can see that the problem of changing hue has been solved, However, as we are normalizing the reconstructed images using minmax normalization, the reconstructed live counterpart of spoof inputs tend to distort as the model starts to converge.

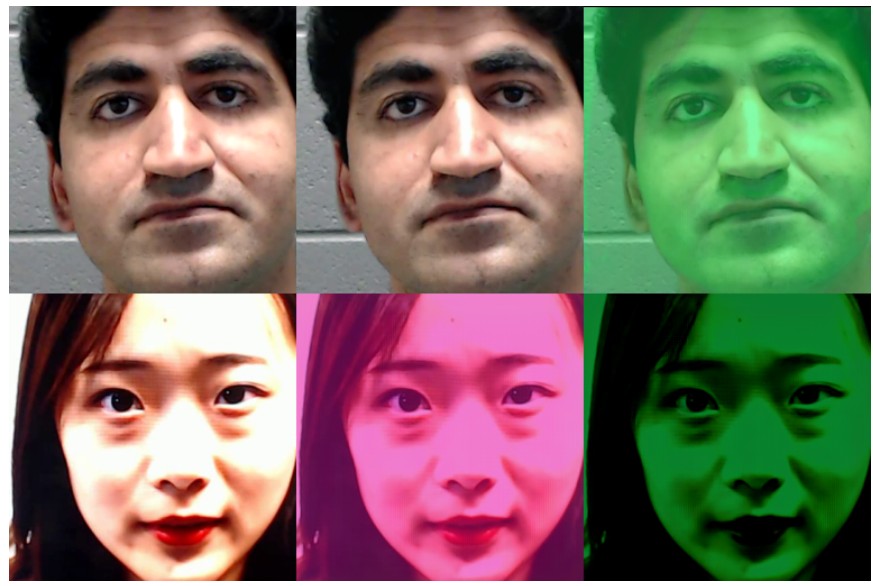

Figure 5: Generated outputs after following the official implemntation. Row-1: (a) Live input, (b) Live counterpart of (a), (c) Spoof counterpart of (a) and, Row-2: (a) Spoof input, (b) Live counterpart of a, (c) Spoof trace of a

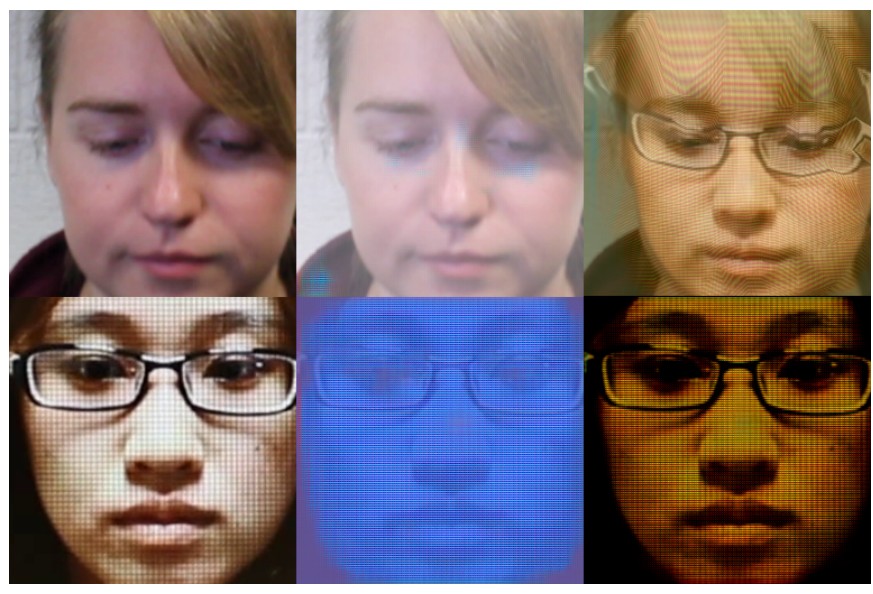

Figure 6: Generated outputs after Minmax Normalization is used. Row-1: (a) Live input, (b) Live counterpart of (a), (c) Spoof counterpart of (a) and, Row-2: (a) Spoof input, (b) Live counterpart of a, (c) Spoof trace of a

### 4.3 Hyperbolic tangent on Spoof-Trace

Although MinMax Normalization solves the problem of unnatural image range partially, another problem arises which is – if we normalize the reconstructed images within $[0, 1]$ range, we might fail to capture the traces where spoof images tend to be darker than live images. To solve this problem, we used equation-5 on the the trace we will get from equation-4. This ensures that the trace will always fall within the limit of $[-1, 1]$ and helps to solve the problem of less bright spoof images.

$$G(I) = I - I_{hat} \tag{4}$$

$$G(I)' = \frac{e^{G(I)} - e^{-G(I)}}{e^{G(I)} + e^{-G(I)}} \tag{5}$$

From Figure-7, we can see that this solution succeeds to find the spoof trace – which is a striped pattern – as well as maintains the color of the image as well. However, like all other above mentioned technique, this also fails to generate perfect outputs as the live counterpart has some jitter in it.

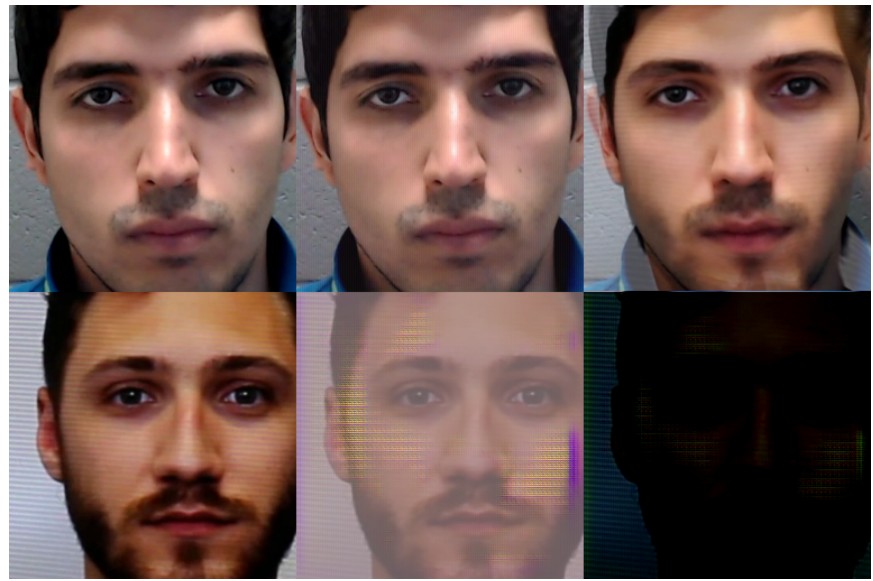

Figure 7: Generated outputs after hyperbolic tangent function is used in the spoof traces. Row-1: (a) Live input, (b) Live counterpart of (a), (c) Spoof counterpart of (a) and, Row-2: (a) Spoof input, (b) Live counterpart of a, (c) Spoof trace of a

## 5 Spoof Detection Results

From Table-1, it can be seen that, our implementation for MSU-SiW Protocol-1 matches their ACER close to $0.53\%$. On the other hand, our implementation on OULU NPU Protocol-1 outperforms their results by almost $0.3\%$ which shows the robustness of our implementation.

| Protocol | Official results | | | Our implementation | | |
|---|---|---|---|---|---|---|
| | APCER | BPCER | ACER | APCER | BPCER | ACER |
| MSU-SiW protocol 1 | 0.0 | 0.0 | 0.0 | 1.0 | 0.05 | 0.53 |
| OULU-NPU protocol 1 | 0.8 | 1.3 | 1.1 | 1.3 | 1.09 | 1.195 |

Table 1: Performance comparison between their and our implementation.

However, as discussed previously, our implementation based on their official repository demonstrates a discolored hue and fails to generate perfect spoof and live images.

## 6 Discussion

During the time of our study of the paper, we have observed many aspects of the paper. Some of these strengthen the paper and some of these are just mere scopes to improve the paper. In the following subsections, we will discuss them.

### 6.1 What worked

Firstly, we would like to discuss our observation of the original paper. We found the paper to match their implementation almost in every cases. The things that did not match are just the improvements that the authors are continuously working hard for.

We found the claim that – live inputs will produce blackened spoof trace and spoof input will highlight the spoof textures – to be true. Also, the claim that outside cues hinder the generalization of face-anti-spoofing to be true as well. For this, we took Fast Fourier Transformed features as an outside cue and the results were not as effective as the original one which we showed in Figure-4.

Secondly, in our extended work, we used $tanh\,function$ on spoof-traces to set the limit within $[-1, 1]$. This made the synthetic spoof images more realistic, images shown in Figure-7.

### 6.2 What did not work

Although, we managed to reproduce close to $0.53\%$, for the original paper, after a certain time of training, the reconstructed live and spoof images tend to be of more Pink and green hue. We believe, this problem existed because of the traces from equation-4 to not being within $[-1, 1]$ range.

Finally, the problem that we could not solve and seemed to be a big problem is that – whenever there is a side face in one of the inputs, the warping function doesn't work properly which is shown in Figure-8. It creates some form of stretching mark. Besides, whenever there are any extraneous object on the face, the spoof trace network considers them as a part of spoof trace. For instance, spectacles or any kind of birth mark in faces [Figure-9].

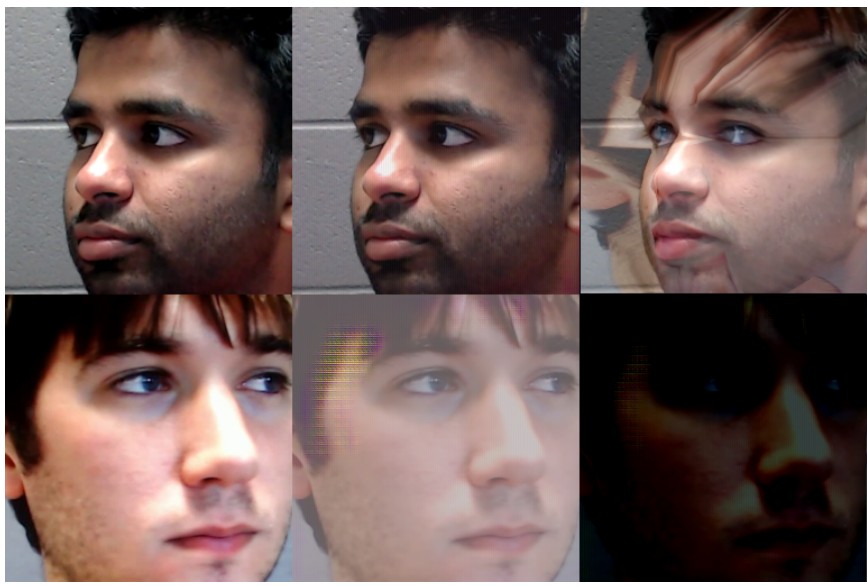

Figure 8: Warping doesn't work well when Sideface is used as an input. Row-1: (a) Live input, (b) Live counterpart of (a), (c) Spoof counterpart of (a) and, Row-2: (a) Spoof input, (b) Live counterpart of a, (c) Spoof trace of a

### 6.3 What was easy

Despite the complicated technique, we found the paper to be understandable because of the diagrams. Moreover, we were able to match most of the techniques with code snippets from the official implementation. This helped us to get a clear perspective of what the author tried to do. Besides, when we found some mismatches – such as: $six$ channel input in the code – we contacted the author and they helped us clear the confusion.

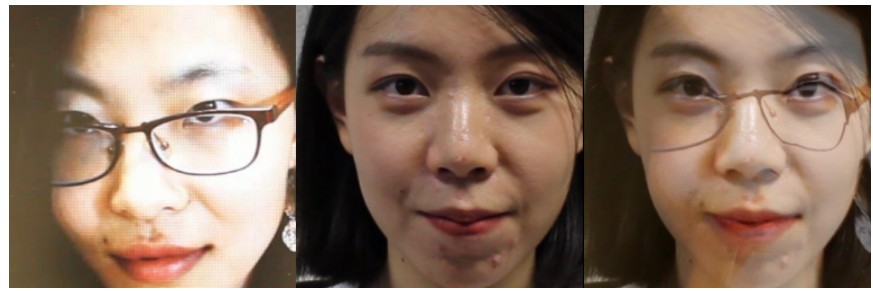

Figure 9: Spectacles are detected as a part of spoof trace. a. Spoof input, b. Live Input, 3. Spoof counterpart of live where spoof trace is from b.

Besides, all the hyperparameters for training the technique were given which saved us from brute-forcing the training hyperparameters.

We advise anyone who wants to re-implement this paper is just to read the code and the paper concurrently and write things up and draw diagrams to make things easier.

### 6.4 What was difficult

To get the metrics, the paper expects us to depend on a lot of hyperparameters. However, these were not explicitly mentioned anywhere in the paper. We believe that this part of the paper can be highly improved to make it less hyperparamer-prone. However, as a temporary solution, we followed their suggestion and used brute-force method to get to the best solution.

Furthermore, the paper is an extremely complicated one to read, not because of its language, but because of the technique. The technique demands to have three training steps and each time with different types of requirements. It made things really difficult for us to re-implement it. To make sure of correctness, we divided the steps into small sections and renamed them in a way that we can identify these sections easily. Later, we verified these separate code snippets to ensure the reproduction. This way we managed to overcome this problem.

### 6.5 Future works

While the original paper introduced us to a briliant approach to tackle face-anti-spoofing, there are several scopes which can be addressed in future:

- The color and textures are not yet up to the mark as the color becomes less vibrant and the texture tends to show artifact effects.
- The warping for the side angle is a scope to work on.
- We can use a self attention so that we can ignore any specific marks on faces.

### 6.6 Communication with original authors

While reading the paper as well as the implementation, we had a few confusions for which we got help from the authors:

1. We couldn't connect the mention of using L2-regularizor in section 3.1 with the implementation. The authors helped us by pointing out to the line number of the code.

2. The authors continuously update the code base with improvements. We found two such inconsistencies we started implementing the code base. The author solved our query and later we integrated the improvements into our codebase.

3. We were unsure how early spoof regressor was working in the code base. The authors ensured us with this as well.

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
