# OpenReview forum: "Reproducibility report for "On Disentangling Spoof Trace forGeneric Face Anti-Spoofing""
_ML_Reproducibility_Challenge/2020 — Reject_

### Official Review · AnonReviewer1 · 2021-02-28
**A good paper but most of the reproduction comes from the official code**

**Rating:** 7
**Confidence:** 3

**Review:**

This paper attempts to reproduce the work published in ECCV 2020, i.e., On Disentangling Spoof Trace for Generic Face Anti-Spoofing. The authors take help from the original paper, the original authors and the official implementation, which took them around a month. They also verified different segments of the original implementation and propose several improvements over a few limitations of the original paper.

Strengths:
The writing is good and easy to follow. The summary of the original paper is clarify and detailed. The authors provide figures to demonstrate whether they have succeeded to reproduce the original results.

Weakness：
1)	As argued in the report, the reproduction took help of the authors and the official implementation. Therefore, it is significant for the authors to put emphases on what are challenging to reproduce the results of the original paper. The FFT adopted in the final decision is confusing. I cannot make it out how it was injected in the original method.
2)	Typos, like the last sentence in the results part in Page 1. It is suggested to go over the paper carefully to improve the quality of the paper.


**Familiar With The Original Paper:**

I have read the original paper

**Reproducibility Summary:**

Report has summary

---

### Official Review · AnonReviewer2 · 2021-03-01
**writing could be improved**

**Rating:** 6
**Confidence:** 5

**Review:**

The study tried to reproduce the paper by following the official implementation. This report can be improved in following aspects:
1. Language could be improved, e.g., line 149 contains typo, some sentences can be shortened.
2. When referring to equations in the original paper, the authors do not explain the context. This makes the study hard to read.
3. The descriptions for figures were not clear to me.

**Familiar With The Original Paper:**

I have not read the original paper

**Reproducibility Summary:**

Report has summary

---

### Official Review · AnonReviewer3 · 2021-03-08
**Review of RR for "On Disentangling Spoof Trace for Generic Face Anti-Spoofing"**

**Rating:** 4
**Confidence:** 4

**Review:**

# Summary and overall assessment

The authors analyzed the reproducibility of "On Disentangling Spoof Trace for Generic Face Anti-Spoofing" which proposes a GAN to identify spoofing in images for robust face recognition. The strengths of this report is that they prototyped the paper ideas from scratch and proposed modifications to improve the methodology. However, the overall report is very vague in writing (see comments below) and I was missing details to understand what has been done, e.g., scope of the reproducibility, description of the approach. Further I do not believe that the contributions to improve the approach is necessarily the scope of a reproducibility report. This can also change the overall results and make them not comparable to the original results.

# Detailed comments on each section

Reproducibility Summary/Scope of reproducibility:
* This is rather a summary of the work itself. I would have expected this paragraph would summarize the content of this report. * "Based on that, this paper suggests": the paper should be citied.
Reproducibility Summary/Results:
* "We succeeded to match the ACER of the original paper to within 0.53%"/ For OULU NPU Protocol-1 the numbers are 1.9% vs. 1.195%, which would result in a difference of 0.705% > 0.53%?
* Last sentence "Later, we proposed a few techniques to" is incomplete.

Introduction:
* "Of all bio-metric authentication technologies, face recognition is the most intuitive and effective." Can you cite the source or elaborate?
* "In the past, hand-crafted features like HOG and LBP were used to tackle the problem of face anti-spoofing." Please cite relevant works.
* "In recent years, CNNs have been adopted as the preferred solution for this problem." Please cite relevant works.
* The introduction could have given more details about solutions to spoofing. Is it a classic classification problem solved with hand-crafted features or CNN features?

Scope of reproducibility:
* Equation 1: Can you explain this equation?

Methodology:
* Can you explain the methodology in details? What is used as input, what problem is optimized, what is the loss function?

Experimental setup:
* What model (architecture) was used here?

Implementation:
* "However, there is a functionality which is not available in PyTorch," Can you be concrete and say which functionality?

Resources:
* "During the implementation phase, we found inconsistencies between the paper and the official implementation." Which ones?
Other:
* "The following section formatting is optional, you can also define sections as you deem fit." I believe this can be removed.

**Familiar With The Original Paper:**

I have not read the original paper

**Reproducibility Summary:**

Report has summary

---

### Decision · Program_Chairs · 2021-03-31

**Decision:**

Reject

**Comment:**

While the contributions to improve the approach are necessarily the scope of a reproducibility report, this work does not carry this out comprehensively.